# ∇*Sim*: Differentiable simulation for system identification and visuomotor control

**https://gradsim.github.io**

Krishna Murthy Jatavallabhula[*1,3,4], Miles Macklin[*2], Florian Golemo[1,3], Vikram Voleti[3,4], Linda Petrini[3], Martin Weiss[3,4], Breandan Considine[3,5], Jérôme Parent-Lévesque[3,5], Kevin Xie[2,6,7], Kenny Erleben[8], Liam Paull[1,3,4], Florian Shkurti[6,7], Derek Nowrouzezahrai[3,5], and Sanja Fidler[2,6,7]

[1]Montreal Robotics and Embodied AI Lab, [2]NVIDIA, [3]Mila, [4]Université de Montréal, [5]McGill, [6]University of Toronto, [7]Vector Institute, [8]University of Copenhagen

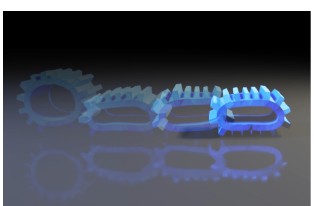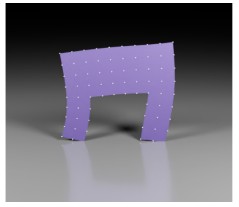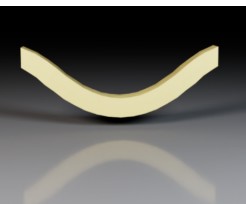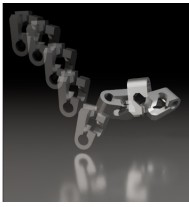

Figure 1: ∇*Sim* is a unified differentiable rendering and multiphysics framework that allows solving a range of control and parameter estimation tasks (rigid bodies, deformable solids, and cloth) directly from images/video.

## Abstract

We consider the problem of estimating an object's physical properties such as mass, friction, and elasticity directly from video sequences. Such a system identification problem is fundamentally ill-posed due to the loss of information during image formation. Current solutions require precise 3D labels which are labor-intensive to gather, and infeasible to create for many systems such as deformable solids or cloth. We present ∇*Sim*, a framework that overcomes the dependence on 3D supervision by leveraging differentiable multiphysics simulation and differentiable rendering to jointly model the evolution of scene dynamics and image formation. This novel combination enables backpropagation from pixels in a video sequence through to the underlying physical attributes that generated them. Moreover, our unified computation graph – spanning from the dynamics and through the rendering process – enables learning in challenging visuomotor control tasks, without relying on state-based (3D) supervision, while obtaining performance competitive to or better than techniques that rely on precise 3D labels.

## 1 Introduction

Accurately predicting the dynamics and physical characteristics of objects from image sequences is a long-standing challenge in computer vision. This end-to-end reasoning task requires a fundamental understanding of *both* the underlying scene dynamics and the imaging process. Imagine watching a short video of a basketball bouncing off the ground and ask: "Can we infer the mass and elasticity of the ball, predict its trajectory, and make informed decisions, e.g., how to pass and shoot?" These seemingly simple questions are extremely challenging to answer even for modern computer vision models. The underlying physical attributes of objects and the system dynamics need to be modeled and estimated, all while accounting for the loss of information during 3D to 2D image formation.

Depending on the assumptions on the scene structre and dynamics, three types of solutions exist: *black*, *grey*, or *white box*. *Black box* methods (Watters et al., 2017; Xu et al., 2019b; Janner et al., 2019; Chang et al., 2016) model the state of a dynamical system (such as the basketball's trajectory in time) as a learned embedding of its states or observations. These methods require few prior assumptions about the system itself, but lack interpretability due to entangled variational factors (Chen et al., 2016) or due to the ambiguities in unsupervised learning (Greydanus et al., 2019; Cranmer et al., 2020b). Recently, *grey box* methods (Mehta et al., 2020) leveraged partial knowledge about the system dynamics to improve performance. In contrast, *white box* methods (Degrave et al., 2016; Liang et al., 2019; Hu et al., 2020; Qiao et al., 2020) impose prior knowledge by employing explicit dynamics models, reducing the space of learnable parameters and improving system interpretability.

---

[*]Equal contribution

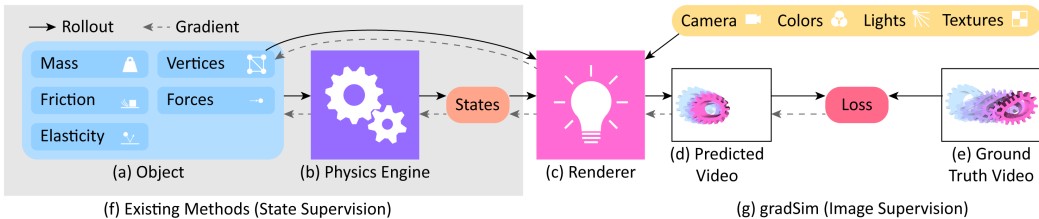

Figure 2: ∇**Sim**: Given video observations of an evolving physical system (e), we randomly initialize scene object properties (a) and evolve them over time using a differentiable physics engine (b), which generates *states*. Our renderer (c) processes states, object vertices and global rendering parameters to produce image frames for computing our loss. We backprop through this computation graph to estimate physical attributes and controls. Existing methods rely solely on differentiable physics engines and require supervision in state-space (f), while ∇*Sim* only needs image-space supervision (g).

Most notably in our context, all of these approaches require precise 3D labels – which are labor-intensive to gather, and infeasible to generate for many systems such as deformable solids or cloth.

**We eliminate the dependence of white box dynamics methods on 3D supervision by coupling explicit (and differentiable) models of scene dynamics with image formation (rendering)**[1].

Explicitly modeling the end-to-end dynamics and image formation underlying video observations is challenging, even with access to the full system state. This problem has been treated in the vision, graphics, and physics communities (Pharr et al., 2016; Macklin et al., 2014), leading to the development of robust forward simulation models and algorithms. These simulators are not readily usable for solving *inverse* problems, due in part to their non-differentiability. As such, applications of black-box *forward* processes often require surrogate gradient estimators such as finite differences or REINFORCE (Williams, 1992) to enable any learning. Likelihood-free inference for black-box forward simulators (Ramos et al., 2019; Cranmer et al., 2020a; Kulkarni et al., 2015; Yildirim et al., 2017; 2015; 2020; Wu et al., 2017b) has led to some improvements here, but remains limited in terms of data efficiency and scalability to high dimensional parameter spaces. Recent progress in *differentiable simulation* further improves the learning dynamics, however we still lack a method for end-to-end differentiation through the entire simulation process (i.e., from video pixels to physical attributes), a prerequisite for effective learning from video frames alone.

We present ∇*Sim*, a versatile end-to-end differentiable simulator that adopts a holistic, unified view of differentiable dynamics and image formation(*cf.* Fig. 1,2). Existing differentiable physics engines only model time-varying dynamics and require supervision in *state space* (usually 3D tracking). We additionally model a differentiable image formation process, thus only requiring target information specified in *image space*. This enables us to backpropagate (Griewank & Walther, 2003) training signals from video pixels all the way to the underlying physical and dynamical attributes of a scene.

Our main contributions are:

- ∇*Sim*, a differentiable simulator that demonstrates the ability to backprop from video pixels to the underlying physical attributes (*cf.* Fig. 2).
- We demonstrate recovering many physical properties exclusively from video observations, including friction, elasticity, deformable material parameters, and visuomotor controls (sans 3D supervision)
- A PyTorch framework facilitating interoperability with existing machine learning modules.

We evaluate ∇*Sim*'s effectiveness on parameter identification tasks for rigid, deformable and thin-shell bodies, and demonstrate performance that is competitive, or in some cases superior, to current physics-only differentiable simulators. Additionally, we demonstrate the effectiveness of the gradients provided by ∇*Sim* on challenging visuomotor control tasks involving deformable solids and cloth.

## 2 RELATED WORK

**Differentiable physics simulators** have seen significant attention and activity, with efforts centered around embedding physics structure into autodifferentiation frameworks. This has enabled differentiation through contact and friction models (Toussaint et al., 2018; de Avila Belbute-Peres et al.,

---

[1]*Dynamics* refers to the laws governing the motion and deformation of objects over time. *Rendering* refers to the interaction of these scene elements – including their material properties – with scene lighting to form image sequences as observed by a virtual camera. *Simulation* refers to a unified treatment of these two processes.

2018; Song & Boularias, 2020b;a; Degrave et al., 2016; Wu et al., 2017a; Research, 2020 (accessed May 15, 2020), latent state models (Guen & Thome, 2020; Schenck & Fox, 2018; Jaques et al., 2020; Heiden et al., 2019), volumetric soft bodies (Hu et al., 2019; 2018; Liang et al., 2019; Hu et al., 2020), as well as particle dynamics (Schenck & Fox, 2018; Li et al., 2019; 2020; Hu et al., 2020). In contrast, $\nabla Sim$ addresses a superset of simulation scenarios, by coupling the physics simulator with a differentiable rendering pipeline. It also supports tetrahedral FEM-based hyperelasticity models to simulate deformable solids and thin-shells.

Recent work on **physics-based deep learning** injects structure in the latent space of the dynamics using Lagrangian and Hamiltonian operators (Greydanus et al., 2019; Chen et al., 2020; Toth et al., 2020; Sanchez-Gonzalez et al., 2019; Cranmer et al., 2020b; Zhong et al., 2020), by explicitly conserving physical quantities, or with ground truth supervision (Asenov et al., 2019; Wu et al., 2016; Xu et al., 2019b).

Sensor readings have been used to predicting the effects of forces applied to an object in models of **learned** (Fragkiadaki et al., 2016; Byravan & Fox, 2017) and **intuitive physics** (Ehsani et al., 2020; Mottaghi et al., 2015; 2016; Gupta et al., 2010; Ehrhardt et al., 2018; Yu et al., 2015; Battaglia et al., 2013; Mann et al., 1997; Innamorati et al., 2019; Standley et al., 2017). This also includes approaches that learn to model multi-object interactions (Watters et al., 2017; Xu et al., 2019b; Janner et al., 2019; Ehrhardt et al., 2017; Chang et al., 2016; Agrawal et al., 2016). In many cases, intuitive physics approaches are limited in their prediction horizon and treatment of complex scenes, as they do not sufficiently accurately model the 3D geometry nor the object properties. **System identification** based on parameterized physics models (Salzmann & Urtasun, 2011; Brubaker et al., 2010; Kozlowski, 1998; Wensing et al., 2018; Brubaker et al., 2009; Bhat et al., 2003; 2002; Liu et al., 2005; Grzeszczuk et al., 1998; Sutanto et al., 2020; Wang et al., 2020; 2018a) and inverse simulation (Murray-Smith, 2000) are closely related areas.

There is a rich literature on **neural image synthesis**, but we focus on methods that model the 3D scene structure, including voxels (Henzler et al., 2019; Paschalidou et al., 2019; Smith et al., 2018b; Nguyen-Phuoc et al., 2018), meshes (Smith et al., 2020; Wang et al., 2018b; Groueix et al., 2018; Alhaija et al., 2018), and implicit shapes (Xu et al., 2019a; Chen & Zhang, 2019; Michalkiewicz et al., 2019; Niemeyer et al., 2020; Park et al., 2019; Mescheder et al., 2019). Generative models condition the rendering process on samples of the 3D geometry (Liao et al., 2019). Latent factors determining 3D structure have also been learned in generative models (Chen et al., 2016; Eslami et al., 2018). Additionally, implicit neural representations that leverage differentiable rendering have been proposed (Mildenhall et al., 2020; 2019) for realistic view synthesis. Many of these representations have become easy to manipulate through software frameworks like Kaolin (Jatavallabhula et al., 2019), Open3D (Zhou et al., 2018), and PyTorch3D (Ravi et al., 2020).

**Differentiable rendering** allows for image gradients to be computed w.r.t. the scene geometry, camera, and lighting inputs. Variants based on the rasterization paradigm (NMR (Kato et al., 2018), OpenDR (Loper & Black, 2014), SoftRas (Liu et al., 2019)) blur the edges of scene triangles prior to image projection to remove discontinuities in the rendering signal. DIB-R (Chen et al., 2019) applies this idea to background pixels and proposes an interpolation-based rasterizer for foreground pixels. More sophisticated differentiable renderers can treat physics-based light transport processes (Li et al., 2018; Nimier-David et al., 2019) by ray tracing, and more readily support higher-order effects such as shadows, secondary light bounces, and global illumination.

## 3 $\nabla Sim$: A UNIFIED DIFFERENTIABLE SIMULATION ENGINE

Typically, physics estimation and rendering have been treated as disjoint, mutually exclusive tasks. In this work, we take on a unified view of *simulation* in general, to compose physics estimation *and* rendering. Formally, simulation is a function $\text{Sim} : \mathbb{R}^P \times [0, 1] \mapsto \mathbb{R}^H \times \mathbb{R}^W; \text{Sim}(\mathbf{p}, t) = \mathcal{I}$. Here $\mathbf{p} \in \mathbb{R}^P$ is a vector representing the simulation state and parameters (objects, their physical properties, their geometries, etc.), $t$ denotes the time of simulation (conveniently reparameterized to be in the interval $[0, 1]$). Given initial conditions $\mathbf{p}_0$, the simulation function produces an image $\mathcal{I}$ of height $H$ and width $W$ at each timestep $t$. If this function Sim were differentiable, then the gradient of $\text{Sim}(\mathbf{p}, t)$ with respect to the simulation parameters $\mathbf{p}$ provides the change in the output of the simulation from $\mathcal{I}$ to $\mathcal{I} + \nabla \text{Sim}(\mathbf{p}, t)\delta\mathbf{p}$ due to an *infinitesimal perturbation of $\mathbf{p}$ by $\delta\mathbf{p}$* . This construct enables a gradient-based optimizer to estimate physical parameters from video , by defining a *loss function* over the image space $\mathcal{L}(\mathcal{I}, .)$, and descending this loss landscape along a

direction parallel to $-\nabla\text{Sim}(.)$ . To realise this, we turn to the paradigms of *computational graphs* and *differentiable programming*.

$\nabla$*Sim* comprises two main components: a *differentiable physics engine* that computes the physical states of the scene at each time instant, and a *differentiable renderer* that renders the scene to a 2D image. Contrary to existing differentiable physics (Toussaint et al., 2018; de Avila Belbute-Peres et al., 2018; Song & Boularias, 2020b;a; Degrave et al., 2016; Wu et al., 2017a; Research, 2020 (accessed May 15, 2020; Hu et al., 2020; Qiao et al., 2020) or differentiable rendering (Loper & Black, 2014; Kato et al., 2018; Liu et al., 2019; Chen et al., 2019) approaches, we adopt a holistic view and construct a computational graph spanning them both.

## 3.1 DIFFERENTIABLE PHYSICS ENGINE

Under Lagrangian mechanics, the state of a physical system can be described in terms of generalized coordinates $\mathbf{q}$, generalized velocities $\dot{\mathbf{q}} = \mathbf{u}$, and design/model parameters $\theta$. For the purpose of exposition, we make no distinction between rigid bodies, or deformable solids, or thin-shell models of cloth, etc. Although the specific choices of coordinates and parameters vary, the simulation procedure is virtually unchanged. We denote the combined state vector by $\mathbf{s}(t) = [\mathbf{q}(t), \mathbf{u}(t)]$.

The dynamic evolution of the system is governed by second order differential equations (ODEs) of the form $\mathbf{M}(\mathbf{s}, \theta)\dot{\mathbf{s}} = \mathbf{f}(\mathbf{s}, \theta)$, where $\mathbf{M}$ is a mass matrix that depends on the state and parameters. The forces on the system may be parameterized by design parameters (e.g. Young's modulus). Solutions to these ODEs may be obtained through black box numerical integration methods, and their derivatives calculated through the continuous adjoint method (Chen et al., 2018). However, we instead consider our physics engine as a differentiable operation that provides an implicit relationship between a state vector $\mathbf{s}^- = \mathbf{s}(t)$ at the start of a time step, and the updated state at the end of the time step $\mathbf{s}^+ = \mathbf{s}(t + \Delta t)$. An arbitrary discrete time integration scheme can be then be abstracted as the function $\mathbf{g}(\mathbf{s}^-, \mathbf{s}^+, \theta) = \mathbf{0}$ , relating the initial and final system state and the model parameters $\theta$ .

Gradients through this dynamical system can be computed by graph-based autodiff frameworks (Paszke et al., 2019; Abadi et al., 2015; Bradbury et al., 2018), or by program transformation approaches (Hu et al., 2020; van Merriënboer et al., 2018). Our framework is agnostic to the specifics of the differentiable physics engine, however in Appendices A through D we detail an efficient approach based on the source-code transformation of parallel kernels, similar to DiffTaichi (Hu et al., 2020). In addition, we describe extensions to this framework to support mesh-based tetrahedral finite-element models (FEMs) for deformable and thin-shell solids. This is important since we require surface meshes to perform differentiable rasterization as described in the following section.

## 3.2 DIFFERENTIABLE RENDERING ENGINE

A renderer expects *scene description* inputs and generates color image outputs, all according to a sequence of image formation stages defined by the *forward* graphics pipeline. The scene description includes a complete *geometric* descriptor of scene elements, their associated material/reflectance properties, light source definitions, and virtual camera parameters. The rendering process is not generally differentiable, as *visibility* and *occlusion* events introduce discontinuities. Most interactive renderers, such as those used in real-time applications, employ a *rasterization* process to project 3D geometric primitives onto 2D pixel coordinates, resolving these visibility events with non-differentiable operations.

Our experiments employ two differentiable alternatives to traditional rasterization, SoftRas (Liu et al., 2019) and DIB-R (Chen et al., 2019), both of which replace discontinuous triangle mesh edges with smooth sigmoids. This has the effect of blurring triangle edges into semi-transparent boundaries, thereby removing the non-differentiable discontinuity of traditional rasterization. DIB-R distinguishes between *foreground pixels* (associated to the principal object being rendered in the scene) and *background pixels* (for all other objects, if any). The latter are rendered using the same technique as SoftRas while the former are rendered by bilinearly sampling a texture using differentiable UV coordinates.

$\nabla$*Sim* performs differentiable physics simulation and rendering at independent and adjustable rates, allowing us to trade computation for accuracy by rendering fewer frames than dynamics updates.

# 4 EXPERIMENTS

We conducted multiple experiments to test the efficacy of $\nabla Sim$ on *physical parameter identification from video* and *visuomotor control*, to address the following questions:

- Can we accurately identify physical parameters by backpropagating from video pixels, through the simulator? (Ans: *Yes, very accurately*, cf. 4.1)
- What is the performance gap associated with using $\nabla Sim$ (2D supervision) vs. differentiable physics-only engines (3D supervision)? (Ans: $\nabla Sim$ *is competitive/superior*, cf. Tables 1, 2, 3)
- How do loss landscapes differ across differentiable simulators ($\nabla Sim$) and their non-differentiable counterparts? (Ans: *Loss landscapes for* $\nabla Sim$ *are smooth*, cf. 4.1.3)
- Can we use $\nabla Sim$ for visuomotor control tasks? (Ans: *Yes, without any 3D supervision*, cf. 4.2)
- How sensitive is $\nabla Sim$ to modeling assumptions at system level? (Ans: *Moderately*, cf. Table 4)

Each of our experiments comprises an *environment* $\mathcal{E}$ that applies a particular set of physical forces and/or constraints, a (differentiable) *loss function* $\mathcal{L}$ that implicitly specifies an objective, and an *initial guess* $\theta_0$ of the physical state of the simulation. The goal is to recover optimal physics parameters $\theta^*$ that minimize $\mathcal{L}$, by backpropagating through the simulator.

## 4.1 PHYSICAL PARAMETER ESTIMATION FROM VIDEO

First, we assess the capabilities of $\nabla Sim$ to accurately identify a variety of physical attributes such as mass, friction, and elasticity from image/video observations. To the best of our knowledge, $\nabla Sim$ is the first study to **jointly** infer such fine-grained parameters from video observations. We also implement a set of competitive baselines that use strictly more information on the task.

### 4.1.1 RIGID BODIES (`RIGID`)

Our first environment–`rigid`–evaluates the accuracy of estimating of physical and material attributes of rigid objects from videos. We curate a dataset of $10000$ simulated videos generated from variations of $14$ objects, comprising primitive shapes such as boxes, cones, cylinders, as well as non-convex shapes from ShapeNet (Chang et al., 2015) and DexNet (Mahler et al., 2017). With uniformly sampled initial dimensions, poses, velocities, and physical properties (density, elasticity, and friction parameters), we apply a *known* impulse to the object and record a video of the resultant trajectory. Inference with $\nabla Sim$ is done by guessing an initial mass (uniformly random in the range $[2, 12]kg/m^3$), unrolling a *differentiable* simulation using this guess, comparing the rendered out video with the true video (pixelwise mean-squared error - MSE), and performing gradient descent updates. We refer the interested reader to the appendix (Sec. G) for more details.

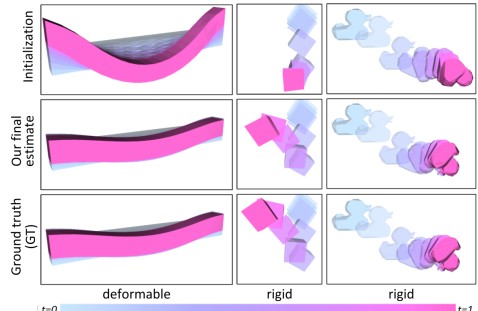

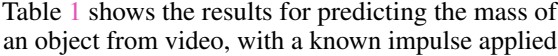

Figure 3: **Parameter Estimation**: For `deformable` experiments, we optimize the material properties of a beam to match a video of a beam hanging under gravity. In the `rigid` experiments, we estimate contact parameters (elasticity/friction) and object density to match a video (GT). We visualize entire time sequences (t) with color-coded blends.

Table 1 shows the results for predicting the mass of an object from video, with a known impulse applied to it. We use EfficientNet (B0) (Tan & Le, 2019) and resize input frames to $64 \times 64$. Feature maps at a resoluition of $4 \times 4 \times 32$ are concatenated for all frames and fed to an MLP with 4 linear layers, and trained with an MSE loss. We compare $\nabla Sim$ with three other baselines: PyBullet + REINFORCE (Ehsani et al., 2020; Wu et al., 2015), diff. physics only (requiring 3D supervision), and a ConvLSTM baseline adopted from Xu et al. (2019b) but with a stronger backbone. The *DiffPhysics* baseline is a strict subset of $\nabla Sim$, it only inolves the differentiable physics engine. However, it needs precise 3D states as supervision, which is the primary factor for its superior performance. Nevertheless, $\nabla Sim$ is able to very precisely estimate mass from video, to a absolute relative error of 9.01e-5, nearly two orders of magnitude better than the ConvLSTM baseline. Two other baselines are also used: the "*Average*" baseline always predicts the dataset mean and the "*Random*" baseline

| Approach | Mean abs. err. (kg) | Abs. Rel. err. |
|---|---|---|
| Average | 0.2022 | 0.1031 |
| Random | 0.2653 | 0.1344 |
| ConvLSTM Xu et al. (2019b) | 0.1347 | 0.0094 |
| PyBullet + REINFORCE Ehsani et al. (2020) | 0.0928 | 0.3668 |
| DiffPhysics (3D sup.) | 1.35e-9 | 5.17e-9 |
| $\nabla Sim$ | 2.36e-5 | 9.01e-5 |

Table 1: **Mass estimation**: $\nabla Sim$ obtains *precise* mass estimates, comparing favourably even with approaches that require 3D supervision (*diffphysics*). We report the mean abolute error and absolute relative errors for all approaches evaluated.

| Approach | mass $m$ | elasticity $k_d$ | elasticity $k_e$ | friction $k_f$ | friction $\mu$ |
|---|---|---|---|---|---|
| Average | 1.7713 | 3.7145 | 2.3410 | 4.1157 | 0.4463 |
| Random | 10.0007 | 4.18 | 2.5454 | 5.0241 | 0.5558 |
| ConvLSTM Xu et al. (2019b) | 0.029 | 0.14 | 0.14 | 0.17 | 0.096 |
| DiffPhysics (3D sup.) | 1.70e-8 | 0.036 | 0.0020 | 0.0007 | 0.0107 |
| $\nabla Sim$ | 2.87e-4 | 0.4 | 0.0026 | 0.0017 | 0.0073 |

Table 2: **Rigid-body parameter estimation**: $\nabla Sim$ estimates contact parameters (elasticity, friction) to a high degree of accuracy, despite estimating them from video. Diffphys. requires accurate 3D ground-truth at 30 FPS. We report absolute *relative* errors for each approach evaluated.

| | Deformable solid FEM | | | Thin-shell (cloth) |
|---|---|---|---|---|
| | Per-particle mass | Material properties | | Per-particle velocity |
| | $m$ | $\mu$ | $\lambda$ | $v$ |
| **Approach** | **Rel. MAE** | **Rel. MAE** | **Rel. MAE** | **Rel. MAE** |
| DiffPhysics (3D Sup.) | 0.032 | 0.0025 | 0.0024 | 0.127 |
| $\nabla Sim$ | 0.048 | 0.0054 | 0.0056 | 0.026 |

Table 3: **Parameter estimation of deformable objects**: We estimate per-particle masses and material properties (for solid def. objects) and per-particle velocities for cloth. In the case of cloth, there is a perceivable performance drop in *diffphysics*, as the center of mass of a cloth is often outside the body, which results in ambiguity.

predicts a random parameter value from the test distribution. All baselines and training details can be found in Sec. H of the appendix.

To investigate whether analytical *differentiability* is required, our PyBullet + REINFORCE baseline applies black-box gradient estimation (Williams, 1992) through a non-differentiable simulator (Coumans & Bai, 2016–2019), similar to Ehsani et al. (2020). We find this baseline particularly sensitive to several simulation parameters, and thus worse-performing. In Table 2, we jointly estimate friction and elasticity parameters of our compliant contact model from video observations alone. The trend is similar to Table 1, and $\nabla Sim$ is able to precisely recover the parameters of the simulation. A few examples can be seen in Fig. 3.

### 4.1.2 DEFORMABLE BODIES (DEFORMABLE)

We conduct a series of experiments to investigate the ability of $\nabla Sim$ to recover physical parameters of deformable solids and thin-shell solids (cloth). Our physical model is parameterized by the per-particle mass, and Lamé elasticity parameters, as described in in Appendix C.1. Fig. 3 illustrates the recovery of the elasticity parameters of a beam hanging under gravity by matching the deformation given by an input video sequence. We found our method is able to accurately recover the parameters of 100 instances of deformable objects (cloth, balls, beams) as reported in Table 3 and Fig. 3.

### 4.1.3 SMOOTHNESS OF THE LOSS LANDSCAPE IN $\nabla Sim$

Since $\nabla Sim$ is a complex combination of differentiable non-linear components, we analyze the loss landscape to verify the validity of gradients through the system. Fig. 4 illustrates the loss landscape when optimizing for the mass of a rigid body when all other physical properties are known.

We examine the image-space mean-squared error (MSE) of a unit-mass cube (1 kg) for a range of initializations (0.1 kg to 5 kg). Notably, the loss landscape of $\nabla Sim$ is well-behaved and conducive to momentum-based optimizers. Applying MSE to the first and last frames of the predicted and true videos provides the best gradients. However, for a naive gradient estimator applied to a non-differentiable simulator (PyBullet + REINFORCE), multiple local minima exist resulting in a very narrow region of convergence. This explains $\nabla Sim$'s superior performance in Tables 1, 2, 3.

### 4.2 VISUOMOTOR CONTROL

To investigate whether the gradients computed by $\nabla Sim$ are meaningful for vision-based tasks, we conduct a range of *visuomotor control* experiments involving the actuation of deformable objects towards a *visual* target pose (a single image). In all cases, we evaluate against *diffphysics*, which uses a goal specification and a reward, both defined over the 3D *state-space*.

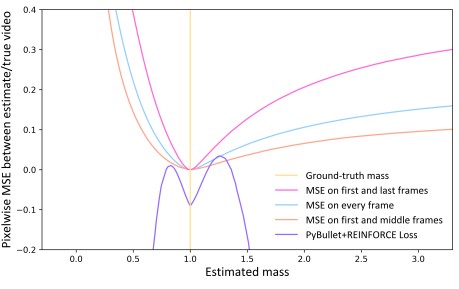 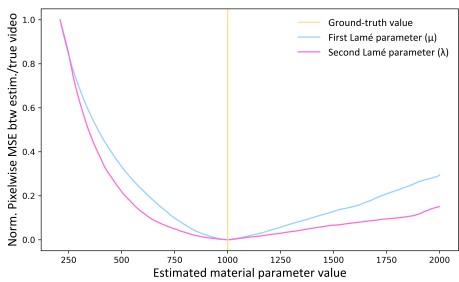

(a) **Loss landscape (rigid)**    (b) **Loss landscape (deformable)**

Figure 4: **Loss landscapes** when optimizing for physical attributes using $\nabla Sim$. (*Left*) When estimating the mass of a rigid-body with known shape using $\nabla Sim$, despite images being formed by a highly nonlinear process (simulation), the loss landscape is remarkably smooth, for a range of initialization errors. (*Right*) when optimizing for the elasicity parameters of a deformable FEM solid. Both the Lamé parameters $\lambda$ and $\mu$ are set to 1000, where the MSE loss has a unique, dominant minimum. Note that, for fair comparison, the ground-truth for our PyBullet+REINFORCE baseline was generated using the PyBullet engine.

### 4.2.1   DEFORMABLE SOLIDS (`CONTROL-WALKER`, `CONTROL-FEM`)

The first example (`control-walker`) involves a 2D walker model. Our goal is to train a neural network (NN) control policy to actuate the walker to reach a target pose on the right-hand side of an image. Our NN consists of one fully connected layer and a `tanh` activation. The network input is a set of 8 time-varying sinusoidal signals, and the output is a scalar activation value per-tetrahedron. $\nabla Sim$ is able to *solve* this environment within three iterations of gradient descent, by minimizing a pixelwise MSE between the last frame of the rendered video and the goal image as shown in Fig. 5 (lower left).

In our second test, we formulate a more challenging 3D control problem (`control-fem`) where the goal is to actuate a soft-body FEM object (a *gear*) consisting of 1152 tetrahedral elements to move to a target position as shown in Fig. 5 (center). We use the same NN architecture as in the 2D walker example, and use the Adam (Kingma & Ba, 2015) optimizer

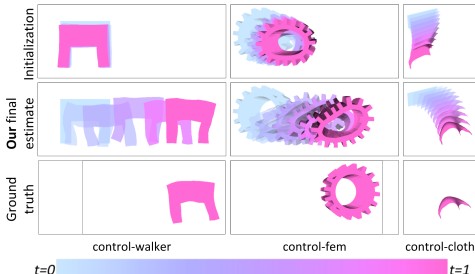

Figure 5: **Visuomotor Control**: $\nabla Sim$ provides gradients suitable for diverse, complex visuomotor control tasks. For `control-fem` and `control-walker` experiments, we train a neural network to actuate a soft body towards a target *image* (GT). For `control-cloth`, we optimize the cloth's initial velocity to hit a target (GT) (specified as an image), under nonlinear lift/drag forces.

to minimize a pixelwise MSE loss. We also train a privileged baseline (*diffphysics*) that uses strong supervision and minimizes the MSE between the target position and the precise 3D location of the center-of-mass (COM) of the FEM model at each time step (i.e. a *dense* reward). We test both *diffphysics* and $\nabla Sim$ against a naive baseline that generates random activations and plot convergence behaviors in Fig. 6a.

While *diffphysics* appears to be a strong performer on this task, it is important to note that it uses explicit 3D supervision at each timestep (i.e. 30 FPS). In contrast, $\nabla Sim$ uses a *single image* as an implicit target, and yet manages to achieve the goal state, albeit taking a longer number of iterations.

### 4.2.2   CLOTH (`CONTROL-CLOTH`)

We design an experiment to control a piece of cloth by optimizing the initial velocity such that it reaches a pre-specified target. In each *episode*, a random cloth is spawned, comprising between 64 and 2048 triangles, and a new start/goal combination is chosen.

In this challenging setup, we notice that *state-based* MPC (*diffphysics*) is often unable to accurately reach the target. We believe this is due to the underdetermined nature of the problem, since, for objects such as cloth, the COM by itself does not uniquely determine the configuration of the object. Visuomotor control on the other hand, provides a more well-defined problem. An illustration of the task is presented in Fig. 5 (column 3), and the convergence of the methods shown in Fig. 6b.

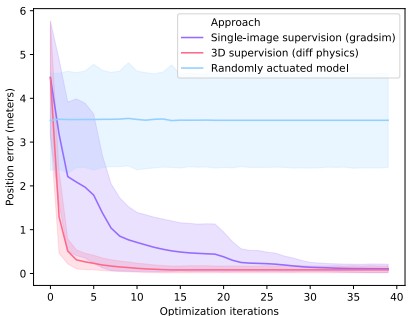 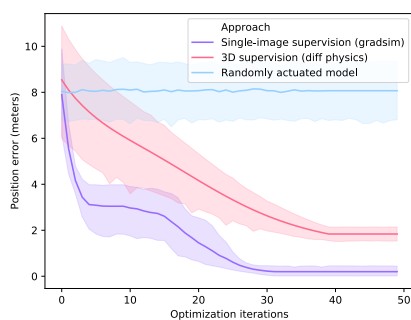

(a) Results of various approaches on the `control-fem` environment (6 randomseeds; each randomseed corresponds to a different goal configuration). While *diffphysics* performs well, it assumes strong 3D supervision. In contrast, ∇*Sim* is able to *solve* the task by using just a *single image* of the target configuration.

(b) Results on `control-cloth` environment (5 randomseeds; each controls the dimensions and initial/target poses of the cloth). *diffphysics* converges to a suboptimal solution due to ambiguity in specifying the pose of a cloth via its center-of-mass. ∇*Sim* *solves* the environment using a single target image.

Figure 6: **Convergence Analysis**: Performance of ∇*Sim* on visuomotor control using image-based supervision, 3D supervision, and random policies.

### 4.3 IMPACT OF IMPERFECT DYNAMICS AND RENDERING MODELS

Being a *white box* method, the performance of ∇*Sim* relies on the choice of dynamics and rendering models employed. An immediate question that arises is "*how would the performance of* ∇Sim *be impacted (if at all) by such modeling choices.*" We conduct multiple experiments targeted at investigating modelling errors and summarize them in Table 4 (left).

We choose a dataset comprising 90 objects equally representing rigid, deformable, and cloth types. By not modeling specific dynamics and rendering phenomena, we create the following 5 variants of our simulator.

1. *Unmodeled friction*: We model all collisions as being frictionless.
2. *Unmodeled elasticity*: We model all collisions as perfectly elastic.
3. *Rigid-as-deformable*: All rigid objects in the dataset are modeled as deformable objects.
4. *Deformable-as-rigid*: All deformable objects in the dataset are modeled as rigid objects.
5. *Photorealistic render*: We employ a photorealistic renderer—as opposed to ∇*Sim*'s differentiable rasterizers—in generating the target images.

In all cases, we evaluate the accuracy with which the mass of the target object is estimated from a target video sequence devoid of modeling discrepancies. In general, we observe that imperfect dynamics models (i.e. unmodeled friction and elasticity, or modeling a rigid object as deformable or vice-versa) have a more profound impact on parameter identification compared to imperfect renderers.

#### 4.3.1 UNMODELED DYNAMICS PHENOMENON

From Table 4 (left), we observe a noticeable performance drop when dynamics effects go unmodeled. Expectedly, the repurcussions of incorrect object type modeling (Rigid-as-deformable, Deformable-as-rigid) are more severe compared to unmodeled contact parameters (friction, elasticity). Modeling a deformable body as a rigid body results in irrecoverable deformation parameters and has the most severe impact on the recovered parameter set.

#### 4.3.2 UNMODELED RENDERING PHENOMENON

We also independently investigate the impact of unmodeled rendering effects (assuming perfect dynamics). We indepenently render ground-truth images and object foreground masks from a photorealistic renderer (Pharr et al., 2016). We use these photorealistic renderings for ground-truth and perform physical parameter estimation from video. We notice that the performance obtained under this setting is superior compared to ones with dynamics model imperfections.

| | Mean Rel. Abs. Err. | | Tetrahedra (#) | Forward (DP) | Forward (DR) | Backward (DP) | Backward (DP + DR) |
|---|---|---|---|---|---|---|---|
| Unmodeled friction | 0.1866 | | 100 | 9057 Hz | 3504 Hz | 3721 Hz | 3057 Hz |
| Unmodeled elasticity | 0.2281 | | 200 | 9057 Hz | 3478 Hz | 3780 Hz | 2963 Hz |
| Rigid-as-deformable | 0.3462 | | 400 | 8751 Hz | 3357 Hz | 3750 Hz | 1360 Hz |
| Deformable-as-rigid | 0.4974 | | 1000 | 4174 Hz | 1690 Hz | 1644 Hz | 1041 Hz |
| Photorealistic render | 0.1793 | | 2000 | 3967 Hz | 1584 Hz | 1655 Hz | 698 Hz |
| Perfect model | **0.1071** | | 5000 | 3871 Hz | 1529 Hz | 1553 Hz | 424 Hz |
| | | | 10000 | 3721 Hz | 1500 Hz | 1429 Hz | 248 Hz |

Table 4: (*Left*) **Impact of imperfect models**: The accuracy of physical parameters estimated by $\nabla Sim$ is impacted by the choice of dynamics and graphics (rendering) models. We find that the system is more sensitive to the choice of dynamics models than to the rendering engine used. (*Right*) **Timing analysis**: We report runtime in simulation steps / second (Hz). $\nabla Sim$ is significantly faster than real-time, even for complex geometries.

### 4.3.3 IMPACT OF SHADING AND TEXTURE CUES

Although our work does not attempt to bridge the reality gap, we show early prototypes to assess phenomena such as shading/texture. Fig. 7 shows the accuracy over time for mass estimation from video. We evaluate three variants of the renderer - "*Only color*", "*Shading*", and "*Texture*". The "*Only color*" variant renders each mesh element in the same color regardless of the position and orientation of the light source. The "*Shading*" variant implements a Phong shading model and can model specular and diffuse reflections. The "*Texture*" variant also applies a non-uniform texture sampled from ShapeNet (Chang et al., 2015). We notice that shading and texture cues significantly improve convergence speed. This is expected, as vertex colors often have very little appearance cues inside the object boundaries, leading to poor correspondences between the rendered and ground-truth images. Furthermore, textures seem to offer slight improvements in convergence speed over shaded models, as highlighted by the inset (log scale) plot in Fig. 7.

### 4.3.4 TIMING ANALYSIS

Table 4 (right) shows simulation rates for the forward and backward passes of each module. We report forward and backward pass rates separately for the differentiable physics (DP) and the differentiable rendering (DR) modules. The time complexity of $\nabla Sim$ is a function of the number of tetrahedrons and/or triangles. We illustrate the arguably more complex case of deformable object simulation for varying numbers of tetrahedra (ranging from 100 to 10000). Even in the case of 10000 tetrahedra—enough to contruct complex mesh models of multiple moving objects—$\nabla Sim$ enables faster-than-real-time simulation (1500 steps/second).

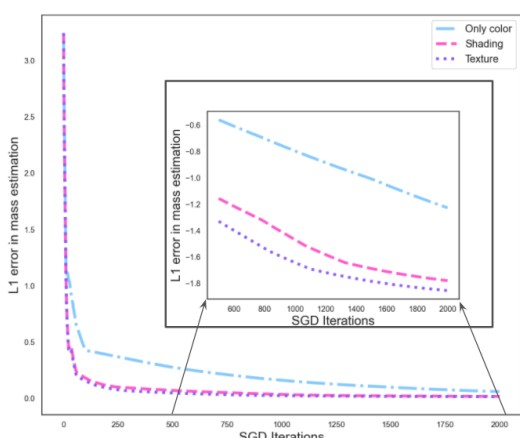

Figure 7: Including **shading and texture cues** lead to faster convergence. Inset plot has a logarithmic Y-axis.

## 5 CONCLUSION

We presented $\nabla Sim$, a versatile differentiable simulator that enables system identification from videos by differentiating through physical processes governing dyanmics and image formation. We demonstrated the benefits of such a holistic approach by estimating physical attributes for time-evolving scenes with complex dynamics and deformations, all from raw video observations. We also demonstrated the applicability of this efficient and accurate estimation scheme on end-to-end visuomotor control tasks. The latter case highlights $\nabla Sim$'s efficient integration with PyTorch, facilitating interoperability with existing machine learning modules. Interesting avenues for future work include extending our differentiable simulation to contact-rich motion, articulated bodies and higher-fidelity physically-based renderers – doing so takes us closer to operating in the real-world.

### ACKNOWLEDGEMENTS

KM and LP thank the IVADO fundamental research project grant for funding. FG thanks CIFAR for project funding under the Catalyst program. FS and LP acknowledge partial support from NSERC.

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

# A APPENDIX

## A DIFFERENTIABLE PHYSICS ENGINE

Under Lagrangian mechanics, the state of a physical system can be described in terms of generalized coordinates $\mathbf{q}$, generalized velocities $\dot{\mathbf{q}} = \mathbf{u}$, and design, or model parameters $\theta$. For the purposes of exposition, we make no distinction between rigid-bodies, deformable solids, or thin-shell models of cloth and other bodies. Although the specific choices of coordinates and parameters vary, the simulation procedure is virtually unchanged. We denote the combined state vector by $\mathbf{s}(t) = [\mathbf{q}(t), \mathbf{u}(t)]$.

The dynamic evolution of the system is governed by a second order differential equations (ODE) of the form $\mathbf{M}\ddot{\mathbf{s}} = \mathbf{f}(\mathbf{s})$, where $\mathbf{M}$ is a mass matrix that may also depend on our state and design parameters $\theta$. Solutions to ODEs of this type may be obtained through black box numerical integration methods, and their derivatives calculated through the continuous adjoint method Chen et al. (2018). However, we instead consider our physics engine as a differentiable operation that provides an implicit relationship between a state vector $\mathbf{s}^- = \mathbf{s}(t)$ at the start of a time step, and the updated state at the end of the time step $\mathbf{s}^+ = \mathbf{s}(t + \Delta t)$. An arbitrary discrete time integration scheme can be then be abstracted as the function $\mathbf{g}(\mathbf{s}^-, \mathbf{s}^+, \theta) = \mathbf{0}$, relating the initial and final system state and the model parameters $\theta$. By the implicit function theorem, if we can specify a loss function $l$ at the output of the simulator, we can compute $\frac{\partial l}{\partial \mathbf{s}^-}$ as $\mathbf{c}^T \frac{\partial \mathbf{g}}{\partial \mathbf{s}^-}$, where $\mathbf{c}$ is the solution to the linear system $\frac{\partial \mathbf{g}}{\partial \mathbf{s}^+}^T \mathbf{c} = -\frac{\partial l}{\partial \mathbf{s}^+}^T$, and likewise for the model parameters $\theta$.

While the partial derivatives $\frac{\partial \mathbf{g}}{\partial \mathbf{s}^-}$, $\frac{\partial \mathbf{g}}{\partial \mathbf{s}^+}$, $\frac{\partial \mathbf{g}}{\partial \theta}$ can be computed by graph-based automatic differentiation frameworks Paszke et al. (2019); Abadi et al. (2015); Bradbury et al. (2018), program transformation approaches such as DiffTaichi, and Google Tangent Hu et al. (2020); van Merriënboer et al. (2018) are particularly well-suited to simulation code. We use an embedded subset of Python syntax, which computes the adjoint of each simulation kernel at runtime, and generates C++/CUDA Kirk et al. (2007) code. Kernels are wrapped as custom autograd operations on PyTorch tensors, which allows users to focus on the definition of physical models, and leverage the PyTorch tape-based autodiff to track the overall program flow. While this formulation is general enough to represent explicit, multi-step, or fully implicit time-integration schemes, we employ semi-implicit Euler integration, which is the preferred integration scheme for most simulators Erez et al. (2015).

### A.1 PHYSICAL MODELS

We now discuss some of the physical models available in $\nabla Sim$.

**Deformable Solids**: In contrast with existing simulators that use grid-based methods for differentiable soft-body simulation Hu et al. (2019; 2020), we adopt a finite element (FEM) model with constant strain tetrahedral elements common in computer graphics Sifakis & Barbic (2012). We use the stable Neo-Hookean constitutive model of Smith et al. Smith et al. (2018a) that derives per-element forces from the following strain energy density:

$$\Psi(\mathbf{q}, \theta) = \frac{\mu}{2}(I_C - 3) + \frac{\lambda}{2}(J - \alpha)^2 - \frac{\mu}{2}\log(I_C + 1), \tag{1}$$

where $I_C, J$ are invariants of strain, $\theta = [\mu, \lambda]$ are the Lamé parameters, and $\alpha$ is a per-element actuation value that allows the element to expand and contract.

Numerically integrating the energy density over each tetrahedral mesh element with volume $V_e$ gives the total elastic potential energy, $U(\mathbf{q}, \theta) = \sum V_e \Psi_e$. The forces due to this potential $\mathbf{f}_e(\mathbf{s}, \theta) = -\nabla_{\mathbf{q}} U(\mathbf{q}, \theta)$, can computed analytically, and their gradients obtained using the adjoint method (*cf.* Section 3.1).

**Deformable Thin-Shells**: To model thin-shells such as clothing, we use constant strain triangular elements embedded in 3D. The Neo-Hookean constitutive model above is applied to model in-plane elastic deformation, with the addition of a bending energy $\mathbf{f}_b(\mathbf{s}, \theta) = k_b \sin(\frac{\phi}{2} + \alpha)\mathbf{d}$, where $k_b$ is the bending stiffness, $\phi$ is the dihedral angle between two triangular faces, $\alpha$ is a per-edge actuation value that allows the mesh to flex inwards or outwards, and $\mathbf{d}$ is the force direction given by Bridson

et al. (2005). We also include a lift/drag model that approximates the effect of the surrounding air on the surface of mesh.

**Rigid Bodies**: We represent the state of a 3D rigid body as $\mathbf{q}_b = [\mathbf{x}, \mathbf{r}]$ consisting of a position $\mathbf{x} \in \mathbb{R}^3$, and a quaternion $\mathbf{r} \in \mathbb{R}^4$. The generalized velocity of a body is $\mathbf{u}_b = [\mathbf{v}, \omega]$ and the dynamics of each body is given by the Newton-Euler equations,

$$\begin{bmatrix} m & \mathbf{0} \\ \mathbf{0} & \mathbf{I} \end{bmatrix} \begin{bmatrix} \dot{\mathbf{v}} \\ \dot{\omega} \end{bmatrix} = \begin{bmatrix} \mathbf{f} \\ \tau \end{bmatrix} - \begin{bmatrix} \mathbf{0} \\ \omega \times \mathbf{I}\omega \end{bmatrix} \tag{2}$$

where the mass $m$ and inertia matrix $\mathbf{I}$ (expressed at the center of mass) are considered design parameters $\theta$.

**Contact**: We adopt a compliant contact model that associates elastic and damping forces with each nodal contact point. The model is parameterized by four scalars $\theta = [k_e, k_d, k_f, \mu]$, corresponding to elastic stiffness, damping, frictional stiffness, and friction coefficient respectively. To prevent interpenetration we use a proportional penalty-based force, $\mathbf{f}_n(\mathbf{s}, \theta) = -\mathbf{n}[k_e C(\mathbf{q}) + k_d \dot{C}(\mathbf{u})]$, where $\mathbf{n}$ is a contact normal, and $C$ is a gap function measure of overlap projected on to $\mathbb{R}^+$. We model friction using a relaxed Coulomb model Todorov (2014) $\mathbf{f}_f(\mathbf{s}, \theta) = -\mathbf{D}[\min(\mu|\mathbf{f}_n|, k_f \mathbf{u}_s)]$, where $\mathbf{D}$ is a basis of the contact plane, and $\mathbf{u}_s = \mathbf{D}^T \mathbf{u}$ is the sliding velocity at the contact point. While these forces are only $C^0$ continuous, we found that this was sufficient for optimization over a variety of objectives.

**More physical simulations**: We also implement a number of other differentiable simulations such as pendula, mass-springs, and incompressible fluids Stam (1999). We note these systems have already been demonstrated in prior art, and thus focus on the more challenging systems in our paper.

## B   DISCRETE ADJOINT METHOD

Above, we presented a formulation of time-integration using the discrete adjoint method that represents an arbitrary time-stepping scheme through the implicit relation,

$$\mathbf{g}(\mathbf{s}^-, \mathbf{s}^+, \theta) = \mathbf{0}. \tag{3}$$

This formulation is general enough to represent both *explicit* or *implicit* time-stepping methods. While explicit methods are often simple to implement, they may require extremely small time-steps for stability, which is problematic for reverse-mode automatic differentiation frameworks that must explicitly store the input state for each discrete timestep invocation of the integration routine. On the other hand, implicit methods can introduce computational overhead or unwanted numerical dissipation Hairer et al. (2006). For this reason, many real-time physics engines employ a semi-implicit (*symplectic*) Euler integration scheme Erez et al. (2015), due to its ease of implementation and numerical stability in most meaningful scenarios (conserves energy for systems where the Hamiltonian is time-invariant).

We now give a concrete example of the discrete adjoint method applied to semi-implicit Euler. For the state variables defined above, the integration step may be written as follows,

$$\mathbf{g}(\mathbf{s}^-, \mathbf{s}^+, \theta) = \begin{bmatrix} \mathbf{u}^+ - \mathbf{u}^- - \Delta t \mathbf{M}^{-1} \mathbf{f}(\mathbf{s}^-) \\ \mathbf{q}^+ - \mathbf{q}^- - \Delta t \mathbf{u}^+ \end{bmatrix} = \mathbf{0}. \tag{4}$$

Note that in general, the mass matrix $\mathbf{M}$ is a function of $\mathbf{q}$ and $\theta$. For conciseness we only consider the dependence on $\theta$, although the overall procedure is unchanged in the general case. We provide a brief sketch of computing the gradients of $\mathbf{g}(\mathbf{s}^-, \mathbf{s}^+, \theta)$. In the case of semi-implicit integration above, these are given by the following equations:

$$\frac{\partial \mathbf{g}}{\partial \mathbf{s}^-} = \begin{bmatrix} -\Delta t \mathbf{M}^{-1} \frac{\partial \mathbf{f}}{\partial \mathbf{q}(t)} & -\mathbf{I} - \Delta t \mathbf{M}^{-1} \frac{\partial \mathbf{f}}{\partial \mathbf{u}(t)} \\ -\mathbf{I} & 0 \end{bmatrix} \quad \frac{\partial \mathbf{g}}{\partial \mathbf{s}^+} = \begin{bmatrix} 0 & \mathbf{I} \\ \mathbf{I} & -\Delta t \mathbf{I} \end{bmatrix} \quad \frac{\partial \mathbf{g}}{\partial \theta} = \begin{bmatrix} -\Delta t \frac{\partial \mathbf{M}^{-1}}{\partial \theta} \\ \mathbf{0} \end{bmatrix}. \tag{5}$$

In the case of semi-implicit Euler, the triangular structure of these Jacobians allows the adjoint variables to be computed explicitly. For fully implicit methods such as backwards Euler, the Jacobians may create a linear system that must be first solved to generate adjoint variables.

## C  PHYSICAL MODELS

We now undertake a more detailed discussion of the physical models implemented in $\nabla Sim$.

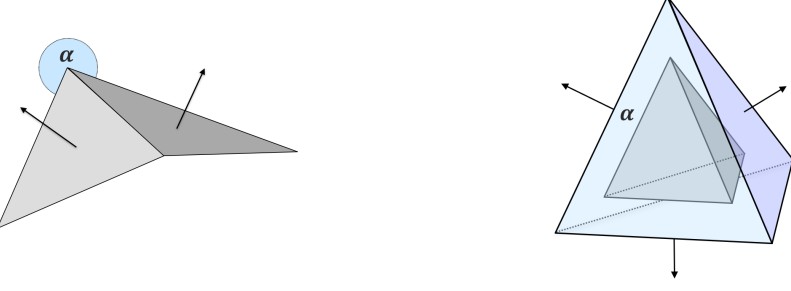

(a) Triangular FEM element        (b) Tetrahedral FEM element

Figure 8: **Mesh Discretization**: We use triangular (a) and tetrahedral (b) FEM models with angle-based and volumetric activation parameters, $\alpha$. These mesh-based discretizations are a natural fit for our differentiable rasterization pipeline, which is designed to operate on triangles.

### C.1  FINITE ELEMENT METHOD

As described in section 3.2 ("Physical models"), we use a hyperelastic constitutive model based on the neo-Hookean model of Smith et al. Smith et al. (2018a):

$$\Psi(\mathbf{q}, \theta) = \frac{\mu}{2}(I_C - 3) + \frac{\lambda}{2}(J - \alpha)^2 - \frac{\mu}{2}\log(I_C + 1). \tag{6}$$

The Lamé parameters, $\lambda, \mu$, control the element's resistance to shearing and volumetric strains. These may be specified on a per-element basis, allowing us to represent heterogeneous materials. In contrast to other work using particle-based models Hu et al. (2020), we adopt a mesh-based discretization for deformable shells and solids. For thin-shells, such as cloth, the surface is represented by a triangle mesh as in Figure 8a, enabling straightforward integration with our triangle mesh-based differentiable rasterizer Liu et al. (2019); Chen et al. (2019). For solids, we use a tetrahedral FEM model as illustrated in Figure 8b. Both these models include a per-element activation parameter $\alpha$, which for thin-shells, allows us to control the relative dihedral angle between two connected faces. For tetrahedral meshes, this enables changing the element's volume, enabling locomotion, as in the `control-fem` example.

### C.2  CONTACT

Implicit contact methods based on linear complementarity formulations (LCP) of contact may be used to maintain hard non-penetration constraints de Avila Belbute-Peres et al. (2018). However, we found relaxed models of contact—used in typical physics engines Erez et al. (2015)—were sufficient for our experiments. In this approach, contact forces are derived from a one-sided quadratic potential, giving rise to penalty forces of the form 9a. While Coulomb friction may also be modeled as an LCP, we use a relaxed model where the *stick* regime is represented by a stiff quadratic potential around the origin, and a linear portion in the *slip* regime, as shown in Figure 9b. To generate contacts, we test each vertex of a mesh against a collision plane and introduce a contact within some distance threshold $d$.

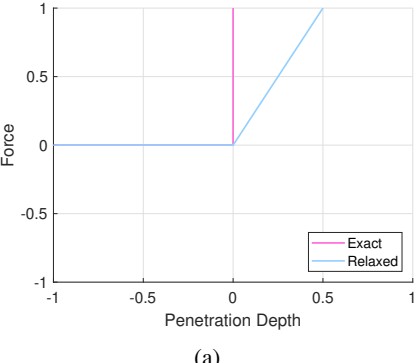 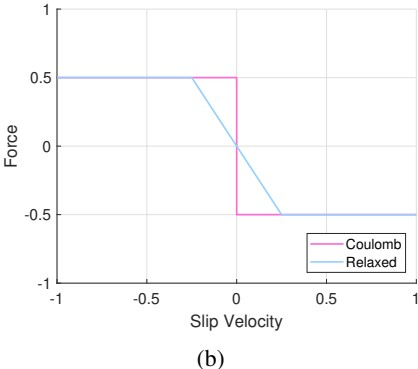

|  |  |
|---|---|
| (a) | (b) |

Figure 9: **Contact Model**: To model non-interpenetration constraints we use a relaxed model of contact that replaces a delta function with a linear hinge corresponding to a quadratic penalty energy (a). To model friction we use a relaxed Coulomb model, that replaces the step function with a symmetric hinge (b).

## C.3   PENDULA

We also implement simple and double pendula, as toy examples of well-behaved and chaotic systems respectively, and estimate the parameters of the system (i.e., the length(s) of the rod(s) and initial angular displacement(s)), by comparing the rendered videos (assuming uniformly random initial guesses) with the true videos. As pendula have extensively been studied in the context of differentiable physics simulation Degrave et al. (2016); de Avila Belbute-Peres et al. (2018); Cranmer et al. (2020b); Toth et al. (2020); Greydanus et al. (2019); Sanchez-Gonzalez et al. (2019), we focus on more challenging systems which have not been studied in prior art.

## C.4   INCOMPRESSIBLE FLUIDS

As an example of incompressible fluid simulation, we implement a smoke simulator following the popular semi-Lagrangian advection scheme of Stam *et al.* Stam (1999). At 2:20 in our supplementary video attachment, we show an experiment which optimizes the initial velocities of smoke particles to form a desired pattern. Similar schemes have already been realized differentiably, e.g. in DiffTaichi Hu et al. (2020) and autograd Maclaurin et al. (2015).

## D   SOURCE-CODE TRANSFORMATION FOR AUTOMATIC DIFFERENTIATION

The discrete adjoint method requires computing gradients of physical quantities with respect to state and design parameters. To do so, we adopt a source code transformation approach to perform reverse mode automatic differentiation Hu et al. (2020); Margossian (2019). We use a domain-specific subset of the Python syntax extended with primitves for representing vectors, matrices, and quaternions. Each type includes functions for acting on them, and the corresponding adjoint method. An example simulation kernel is then defined as follows:

```
1  @kernel
2  def integrate_particles(
3      x : tensor(float3),
4      v : tensor(float3),
5      f : tensor(float3),
6      w : tensor(float),
7      gravity : tensor(float3),
8      dt : float,
9      x_new : tensor(float3),
10     v_new : tensor(float3)
11 ):
12
13     # Get thread ID
14     thread_id = tid()
15
16     # Load state variables and parameters
17     x0 = load(x, thread_id)
```

```
18    v0 = load(v, thread_id)
19    f0 = load(f, thread_id)
20    inv_mass = load(w, thread_id)
21
22    # Load external forces
23    g = load(gravity, 0)
24
25    # Semi-implicit Euler
26    v1 = v0 + (f0 * inv_mass - g * step(inv_mass)) * dt
27    x1 = x0 + v1 * dt
28
29    # Store results
30    store(x_new, thread_id, x1)
31    store(v_new, thread_id, v1)
```

Listing 1: Particle Integration Kernel

At runtime, the kernel's abstract syntax tree (AST) is parsed using Python's built-in `ast` module. We then generate C++ kernel code for forward and reverse mode, which may be compiled to a CPU or GPU executable using the PyTorch `torch.utils.cpp_extension` mechanism.

This approach allows writing imperative code, with fine-grained indexing and implicit operator fusion (since all operations in a kernel execute as one GPU kernel launch). Each kernel is wrapped as a PyTorch autograd operation so that it fits natively into the larger computational graph.

## E   MPC CONTROLLER ARCHITECTURE

For our model predictive control examples, we use a simple 3-layer neural network architecture illustrated in Figure 10. With simulation time $t$ as input we generate $N$ phase-shifted sinusoidal signals which are passed to a fully-connected layer (zero-bias), and a final activation layer. The output is a vector of per-element activation values as described in the previous section.

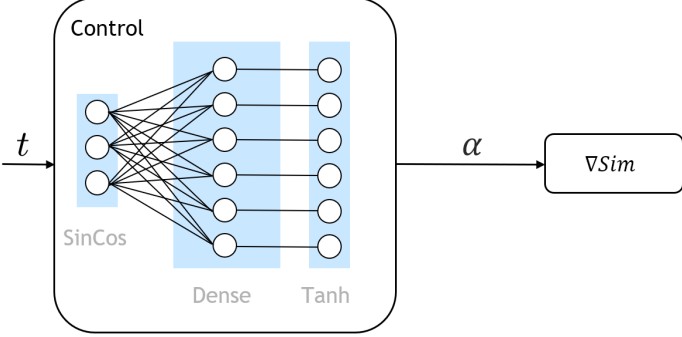

Figure 10: Our simple network architecture used the for `control-walker` and `control-fem` tasks.

## F   LOSS LANDSCAPES FOR PARAMETER ESTIMATION OF DEFORMABLE SOLIDS

$\nabla Sim$ integrates several functional blocks, many of which contain nonlinear operations. Furthermore, we employ a pixelwise mean-squared error (MSE) loss function for estimating physical parameters from video. To demonstrate whether the gradients obtained from $\nabla Sim$ are relevant for the task of physical parameter estimation, in Figure 2 of the main paper, we present an analysis of the MSE loss landscape for mass estimation.

### F.1   ELASTICITY PARAMETER

We now present a similar analysis for elasticity parameter estimation in deformable solids. Figure 11a shows the loss landscape when optimizing for the Lamé parameters of a deformable solid FEM. In this case, both parameters $\lambda$ and $\mu$ are set to 1000. As can be seen in the plot, the loss landscape has

a unique, dominant minimum at 1000. We believe the well-behaved nature of our loss landscape is a key contributing factor to the precise physical-parameter estimation ability of $\nabla Sim$.

### F.2 LOSS LANDSCAPE IN PYBULLET (REINFORCE)

Figure 11 shows how optimization using REINFORCE can introduce complications. As the simulation becomes unstable with masses close to zero, poor local optimum can arise near the mean of the current estimated mass. This illustrates that optimization through REINFORCE is only possible after careful tuning of step size, sampling noise and sampling range. This reduces the utility of this method in a realistic setting where these hyperparameters are not known a priori.

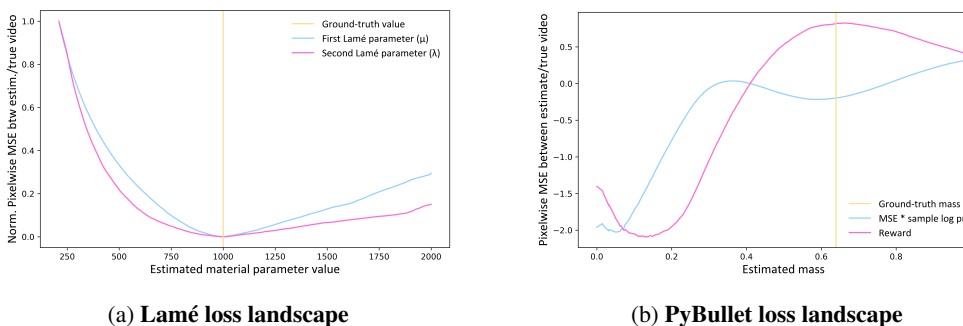

(a) **Lamé loss landscape**      (b) **PyBullet loss landscape**

Figure 11: **Loss Landscapes**: (left) when optimizing for the elasicity parameters of a deformable FEM solid. Both the Lamé parameters $\lambda$ and $\mu$ are set to 1000, where the MSE loss has a unique, dominant minimum. (right) when optimizing for the mass, the reward (negative normalized MSE) has a maximum close to the ground truth maximum but the negative log likelihood of each mass sample that's multiplied with the reward only shows a local minimum that's sensitive to the center of the current mass estimate.

### F.3 IMPACT OF THE LENGTH OF A VIDEO SEQUENCE

To assess the impact of the length of a video on the quality of our solution, we plot the loss landscapes for videos of varying lengths in Fig. 12. We find that shorter videos tend to have steeper loss landscapes compared to longer ones. The frame-rate also has an impact on the steepness of the landscape. In all cases though, the loss landscape is smooth and has the same unique minimum.

## G   DATASET DETAILS

For the rigid-body task of physical parameter estimation from video, we curated a dataset comprising of 14 meshes, as shown in Fig. 13. The objects include a combination of primitive shapes, fruits and vegetables, animals, office objects, and airplanes. For each experiment, we select an object at random, and sample its physical attributes from a predefined range: densities from the range $[2, 12]\ kg/m^3$, contact parameters $k_e, k_d, k_f$ from the range $[1, 500]$, and a coefficient of friction $\mu$ from the range $[0.2, 1.0]$. The positions, orientations, (anisotropic) scale factors, and initial velocities are sampled uniformly at random from a cube of side-length $13m$ centered on the camera. Across all rigid-body experiments, we use 800 objects for training and 200 objects for testing.

## H   BASELINES

In this section, we present implementation details of the baselines used in our experiments.

### H.1   PYBULLET + REINFORCE

To explore whether existing non-differentiable simulators can be employed for physical parameter estimation, we take PyBullet Coumans & Bai (2016–2019) – a popular physics engine – and make it trivially differentiable, by gradient estimation. We employ the REINFORCE Williams (1992)

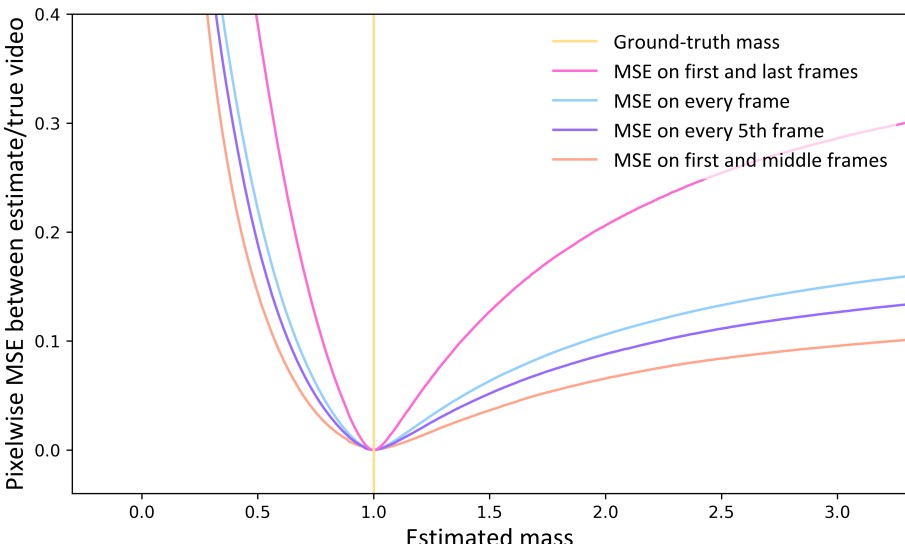

Figure 12: **Impact of the length of a video sequence** on the loss landscape. Notice how the loss landscape is much steeper for smaller videos (e.g., MSE of first and last frames). Nonetheless, all cases have a smooth loss landscape with the same unique minimum.

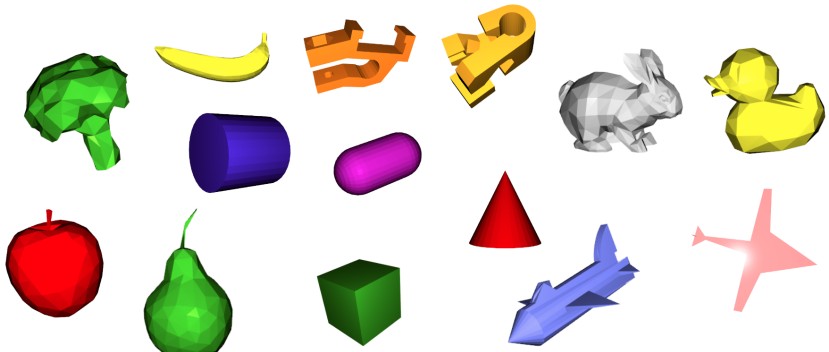

Figure 13: **Objects used** in our rigid-body experiments. All of these meshes have been simplified to contain 250 or fewer vertices, for faster collision detection times.

technique to acquire an approximate gradient through the otherwise non-differentiable environment. The implementation was inspired by Wu et al. (2015) and Rezende et al. (2016). In concurrent work, a similar idea was explored in Ehsani et al. (2020).

In PyBullet, the mass parameter of the object is randomly initialized in the range $[0, N_v]$, where $N_v$ is the number of vertices, the object is set to the same starting position and orientation as in the dataset, and the camera parameters are identical to those used in the dataset. This configuration ensures that if the mass was correct, the video frames rendered out by PyBullet would perfectly align with those generated by $\nabla Sim$. Each episode is rolled out for the same duration as in the dataset (60 frames, corresponding to 2 seconds of motion). In PyBullet this is achieved by running the simulation at 240 Hz and skipping 7 frames between observations. The REINFORCE reward is calculated by summing the individual $L2$ losses between ground truth frames and PyBullet frames, then multiplying each by $-1$ to establish a global maximum at the correct mass, in contrast with a global minimum as in $\nabla Sim$. When all individual frame rewards have been calculated, all trajectory rewards are normalized before calculating the loss. This ensures that the reward is scaled correctly with respect to REINFORCE's negative sample log likelihood, but when the mass value approaches the local optimum, this leads to instability in the optimization process. To mitigate this instability,

we introduce reward decay, which a hyperparameter that slowly decreases the reward values as optimization progresses, in a similar manner to learning rate decay. Before each optimization step, all normalized frame reward values are multiplied by $reward\_decay$. After the optimization step, the decay is updated by $reward\_decay = reward\_decay * decay\_factor$. The hyperparameters used in this baseline can be found in Table 5.

| Parameter | Value | Meaning |
|---|---|---|
| `no_samples` | 5 | How often was the mass sampled at every step |
| `optimization_steps` | 125 | Total number of optimization steps |
| `sample_noise` | 0.05 | Std. dev. of normal distribution that mass is sampled from |
| `decay_factor` | 0.925 | Factor that reward decay is multiplied with after optimizer step |
| `dataset_size` | 200 | Number of bodies that the method was evaluated on |

Table 5: PyBullet-REINFORCE hyperparameters.

### H.2 CNN FOR DIRECT PARAMETER ESTIMATION

In the rigid-body parameter estimation experiments, we train a ConvNet baseline, building on the EfficientNet-B0 architecture Tan & Le (2019). The ConvNet consists of two convolutional layers with parameters (PyTorch convention): $(1280, 128, 1)$, $(128, 32, 1)$, followed by linear layers and ReLU activations with sizes $[7680, 1024, 100, 100, 100, 5]$. No activation is applied over the output of the ConvNet. We train the model to minimize the mean-squared error between the estimated and the true parameters, and use the Adam optimizer Kingma & Ba (2015) with learning rate of $0.0001$. Each model was trained for 100 epochs on a $V100$ GPU. The input image frames were preprocessed by resizing them to $64 \times 64$ pixels (to reduce GPU memory consumption) and the features were extracted with a pretrained EfficientNet-B0.

### I COMPUTE AND TIMING DETAILS

Most of the models presented in $\nabla Sim$ can be trained and evaluated on modern laptops equipped with graphics processing units (GPUs). We find that, on a laptop with an Intel i7 processor and a GeForce GTX 1060 GPU, parameter estimation experiments for rigid/nonrigid bodies can be run in under 5-20 minutes per object on CPU and in under 1 minute on the GPU. The visuomotor control experiments (`control-fem`, `control-cloth`) take about 30 minutes per episode on the CPU and under 5 minutes per episode on the GPU.

### J OVERVIEW OF AVAILABLE DIFFERENTIABLE SIMULATIONS

Table 6 presents an overview of the differentiable simulations implemented in $\nabla Sim$, and the optimizable parameters therein.

| | pos | vel | mass | rot | rest | stiff | damp | actuation | g | $\mu$ | e | ext forces |
|---|---|---|---|---|---|---|---|---|---|---|---|---|
| Rigid body | ✓ | ✓ | ✓ | ✓ | | | | | ✓ | ✓ | ✓ | |
| Simple pendulum | ✓ | | | | | | | | ✓ | | | ✓ |
| Double pendulum | ✓ | | | | | | | | ✓ | | | ✓ |
| Deformable object | ✓ | ✓ | ✓ | ✓ | | | | ✓ | ✓ | ✓ | ✓ | |
| Cloth | ✓ | ✓ | ✓ | | ✓ | ✓ | ✓ | | ✓ | | | |
| Fluid (Smoke) (2D) | | ✓ | | | | | | | | | | |

Table 6: An overview of **optimizable parameters** in $\nabla Sim$. Table columns are (in order, from left to right): Initial particle positions (pos), Initial particle velocities (vel), Per-particle mass (mass), Initial object orientation (rot), Spring rest lengths (rest), Spring stiffnesses (stiff), Spring damping coefficients (damp), Actuation parameters (actuation), Gravity (g), Friction parameters $\mu$, Elasticity parameters (e), External force parameters (ext forces).

## K    LIMITATIONS

While providing a wide range of previously inaccessible capabilities, $\nabla Sim$ has a few limitations that we discuss in this section. These shortcomings also form interesting avenues for subsequent research.

- $\nabla Sim$ (and equivalently $\nabla$PyBullet) are inept at handling **tiny masses** ($100g$ and less). Optimizing for physical parameters for such objects requires a closer look at the design of physics engine and possibly, numerical stability.
- **Articulated bodies** are not currently implemented in $\nabla Sim$. Typically, articulated bodies are composed of multiple prismatic joints which lend additional degrees of freedom to the system.
- While capable of modeling contacts with simple geometries (such as between arbitrary triangle meshes and planar surfaces), $\nabla Sim$ has limited capability to handle **contact-rich** motion that introduces a large number of discontinuities. One way to handle contacts differentiably could be to employ more sophisticated contact detection techniques and solve a *linear complementarity problem* (LCP) at each step, as done in de Avila Belbute-Peres et al. (2018).
- Aside from the aforementioned drawbacks, we note that physics engines are adept at modeling phenomena which can be codified. However, there are several **unmodeled physical phenomena** that occur in **real-world** videos which must be studied in order for $\nabla Sim$ to evolve as a scalable framework capable of operating in the wild.

## L    BROADER IMPACT

Much progress has been made on end-to-end learning in visual domains. If successful, image and video understanding promises far-reaching applications from safer autonomous vehicles to more realistic computer graphics, but relying on these tools for planning and control poses substantial risk.

Neural information processing systems have shown experimentally promising results on visuomotor tasks, yet fail in unpredictable and unintuitive ways when deployed in real-world applications. If embodied learning agents are to play a broader role in the physical world, they must be held to a higher standard of interpretability. Establishing trust requires not just empirical, but explanatory evidence in the form of physically grounded models.

Our work provides a bridge between gradient- and model-based optimization. Explicitly modeling visual dynamics using well-understood physical principles has important advantages for human explainability and debuggability.

Unlike end-to-end neural architectures which distribute bias across a large set of parameters, $\nabla$Sim trades their flexibility for physical interpretability. This does not eliminate the risk of bias in simulation, but allows us to isolate bias to physically grounded variables. Where discrepancy occurs, users can probe the model to obtain end-to-end gradients with respect to variation in physical orientation and material properties, or pixelwise differences. Differentiable simulators like $\nabla$Sim afford a number of opportunities for use and abuse. We envision the following scenarios.

- A technician could query a trained model, "What physical parameters is the steering controller most sensitive to?", or "What happens if friction were slightly lower on that stretch of roadway?"
- An energy-conscious organization could use $\nabla$Sim to accelerate convergence of reinforcement learning models, reducing the energy consumption required for training.
- Using differentiable simulation, an adversary could efficiently construct a physically plausible scene causing the model to produce an incorrect prediction or take an unsafe action.

Video understanding is a world-building exercise with inherent modeling bias. Using physically well-studied models makes those modeling choices explicit, however mitigating the risk of bias still requires active human participation in the modeling process. While a growing number of physically-based rendering and animation efforts are currently underway, our approach does require a high upfront engineering cost in simulation infrastructure. To operationalize these tools, we anticipate practitioners will need to devote significant effort to identifying and replicating unmodeled dynamics from real world-trajectories. Differentiable simulation offers a computationally tractable and physically interpretable pathway for doing so, allowing users to estimate physical trajectories and the properties which govern them.

