# OpenReview forum: "gradSim: Differentiable simulation for system identification and visuomotor control"
_ICLR.cc/2021/Conference — ICLR 2021 Poster_

### Official Review · AnonReviewer3 · 2020-10-27
**gradSim: Differentiable simulation for system identification and visuomotor control**

**Rating:** 7
**Confidence:** 3

**Review:**

This work focuses on the problem of estimating object physical properties from video sequences.

The proposed framework combines differentiable physical simulations and differentiable rendering to map physical parameters into images differentiably. This paradigm is then used to recover physical parameters from image sequences by means of gradient based optimisation.

Validation of the proposed method is carried through two main synthetic applications, parameter identification and visuomotor control.

Although the proposed approach still requires 3D ground truth information to yield reliable estimates, it is and encouraging step towards unsupervised physics understanding from image/video data.

Positive:

-Crucially and differently from previous attempts, the proposed approach does not require 3D supervision - except for geometry and appearance of the static scene (i.e. at t=0).

-Approach is clever, simple and yields interpretable representation

-First step towards physics understanding from videos

Negative:

- I would improve the quality of the visualisations and plots in the paper (e.g. I found Figure 6 impossible to read)

-  How to differentiate through the physical simulator was not obvious to me. I would have appreciated a more detailed explanation of how that is done in practice for one of the physical problems studied in the paper to be included in the main manuscript, in an effort to make the paper more readable.

---

> ### Author Response · Authors · 2020-11-24
> **Thank you for the feedback**
>
> Thank you for your feedback, We have revised our manuscript to address your concerns.
>
>
> **Comment: "I would improve the quality ... plots in the paper ..."**
>
> Response: We have added a summary figure to better ground the applications presented in our paper. We have revised _Figure 6_ (now numbered as _Figure 7_) and added a more comprehensive explanation to the manuscript. For a cleaner demonstration, we exclusively focused on 3 variants, as opposed to 7 in the earlier version.
> We also attempted creating hi-res photorealistic timelapse versions of _Figure 3_ and _Figure 5_, but noticed that the outputs were visually cluttered and hard to interpret. We therefore retained original versions of these figures, and intend to use the photorealistic variants in our video abstract.
>
>
> **Comment: "How to differentiate through the physical simulator was not obvious ..."**
>
> Response: We thank you for raising this concern. This was by design, as our primary objective was the unification of differentiable physics and rendering. The precise details of our simulation framework are deferred to the Appendix (Sections A through E). We have added a note to this effect in the main paper. We hope the release of our code (including data and pretrained models, upon acceptance) will help to alleviate this concern.

---

### Official Review · AnonReviewer4 · 2020-10-28
**Interesting research direction that will spur follow up work despite preliminary nature of evaluation**

**Rating:** 7
**Confidence:** 4

**Review:**

This work presents a fully differentiable physics simulation coupled with neural rendering such that input video can be used to estimate object properties or find control policies to move those objects by trying to generate the same video at the output.

The paper is well motivated by presenting a natural progression of ideas from this literature and it does a thorough job discussing related work. The paper is light on details in section 3 and it is necessary to refer to the appendix to get a complete picture. Overall, the technical contribution is solid and thus worth accepting the paper even if the validation is with relatively simpler experiments since they are sufficient to motivate this direction to be further researched. Below are a few comments to aid in improving the current work:

- All experiments use what I am guessing are input (desired) videos from the same pipeline and then later hiding some parameters (to be learned). While this is a good validation the learning done here is still 'in distribution'. It would be useful to see if video (even simplistic) from a different simulator or simplified from a real world video could be applied. To what extent is this possible and are there any fundamental limitations that prevent this at the moment?

- Analysis is mostly with one object in an empty scene. Are there technical limitations to handling realistic scenes where there are multiple objects and those objects interact with each other as well the environment? How does this affect performance wrt forward and backward pass timings? With such experiments, it would be helpful to understand if the released code can be easily extended to such (more complex) settings or if someone would need to start a new implementation from scratch.

- The scale on the loss landscape is quite small, '0.4 pixelwise mse'. How good does the initial guess need to be to stay in the range, do the curves in fig 3 continue the trend beyond these values for larger error?

- Reality gap: while this is discuss in reference to visual appearances, since the current experiments deal with synthetic scenes, the more relevant topic to discuss is the reality gap wrt physics and object motions. Experiments designed to study this would boost confidence in this approach.


Other comments:

- How much does the performance depend on good initial guess?

- Currently a single impulse is used to set things in motion, can this be extended to handle more continuous actions?

- Presenting qualitative results for baselines would be helpful

- Some baselines not clearly explained: average, random, ConvLSTM

- How does performance scale with the length of the video?

---

> ### Author Response · Authors · 2020-11-24
> **Thank you for the insightful review**
>
> Thank you for your review. Based on your comments, we have significantly revised the presentation of our results and ablation studies.
>
> **Comment: "All experiments ... use ... videos from the same pipeline ... It would be useful to see if video ... from a different simulator ..."**
>
> Response: We have added a more thorough analysis of our experiments with unmodeled dynamics and rendering effects. We believe these experiments help characterize 'out-of-distribution' performance. Furthermore, we investigate the effect of purely relying on the dynamics (under a fixed differentiable renderer) and, likewise, the impact of differentiable rendering cues under a fixed differentiable dynamics model. We find (as reported in _Section 4.3.1_) that dynamics cues tend to have a larger impact on performance compared to rendering cues. We also investigate several other phenomena, including "what happens if contact goes unmodelled?", and "what if a deformable object is accidentally modeled as a rigid object? (and vice versa)". Our experiments include results on photorealistic videos (albeit under perfect object delineation/segmentation), although we agree that an extension to real-world images is the next hurdle to overcome.
>
> **Comment: "Analysis is mostly with one object in an empty scene ..."**
>
> Response: This is largely due to our system identification underpinnings, where we typically estimate the physical attributes of a single object (with known geometry). To the best of our knowledge, there are no technical limitations to handling multiple objects interacting with each other, however it may be the case that multi-object scenes might impose larger sensitivity to initial conditions due to a potentially larger number of contacts.
>
>
> **Comment: "Analysis wrt forward/backward timings ..."**
>
> Response: Thank you for pointing out this detail. Our timing trends depend on the number of tetrahedra/triangles since we currently match each pair of plausible contacts to detect collisions. While this allows for dense differentiability (gradients are available w.r.t. each element of the object geometry), for scalability one might look at ideas similar to those proposed in a recent paper titled "Scalable differentiable physics" [ICML 2020]. It is worthwhile noting that we achieve a forward pass frequency of 3721 Hz and backward pass frequency of 248 Hz when 10,000 tetrahedra are present -- large enough to handle multiple objects with fine-grained details.
>
>
> **Comment: "Reality gap ..."**
>
> Response: We have revised our draft, including a thorough revision of _Section 4.3_ which details experiments to address unmodeled dynamics and rendering effects. As stated in response to an earlier comment, we assess the impact of unmodeled dynamics as well as unmodeled shading. Moreover, _Table 2_ presents an assessment of unmodelled geometry (shape) -- another important real-world attribute.
>
>
> **Comment: "The scale on the loss landscape is quite small ... How good does the initial guess need to be ..."**
>
> Response: We pick a mass uniformly randomly in our operating range of mass densities ([2, 12] kg/m^3). While originally featured in the appendix (_Sec. G_), we have now highlighted this aspect in the main paper (_Sec. 4.1.1_). We also emphasize that our loss landscape is smooth for a wide range of initialization errors (cf. _Fig. 4_). Thank you for raising this issue.
>
>
> **Comment: "Currently a single impulse is used ... extended to handle more continuous actions?"**
>
> Response: We indeed support continuous actions in all our experiments. While an impulse seemed to be the best choice for rigid bodies, all our cloth and deformable solid experiments use continuous (time-varying) actions.
>
>
> **Comment: "Presenting qualitative results for baselines would be helpful."**
>
> Response: Thank you for this suggestion. We believe this is best showcased in a video format and plan to incorporate this in our revised video abstract.
>
>
> **Comment: "Some baselines not clearly explained ..."**
>
> Response: We’ve updated the paper to include more details about the baselines. Additionally, we have referred readers to the appendices (_Sections G and H_) where we explain our baselines, parameters, and training details.
>
>
> **Comment: "How does performance scale to the length of the video ..."**
>
> Response: We have included an additional analysis to this effect in our appendix (_Section F.3._ and _Fig. 12_). We observe that the loss landscape is much steeper for smaller videos than longer ones, indicating longer convergence times. In all cases, it is very smooth and has the same global minimum.
>
>
> We found your feedback both insightful and instrumental in revising our manuscript. Thank you.

---

### Official Review · AnonReviewer1 · 2020-10-29

**Rating:** 7
**Confidence:** 3

**Review:**

- Summary

This paper presents a framework for performing both differentiable physics simulations and differentiable rendering. This fully differentiable simulation and rendering pipeline is then employed to perform system identification tasks, directly from video frames, being able to match or outperform both visual-based and state-based baselines. Moreover, the potential of this framework to be applied for visuomotor control is also demonstrated.


- Pros

This method unified advances in the differentiation of both physics simulation and rendering.

The experimental results demonstrate a good ability to perform system identification for diverse parameters and control directly from videos.
The ability to identify parameters or direct control tasks directly from images is useful, since it reduces the need for direct supervision/annotation in the form of state information.

The presented simulator supports a variety of "domains", such as rigid and deformable body dynamics, cloth simulation, and these are efficient enough to be run faster than real time (at least for simple tasks).



- Cons

Overall, the proposed method is mostly a unification of pre-existing techniques from different fields, such as differentiable rigid and deformable body dynamics, differentiable rendering.

The paper itself admits that a limitation of this method is that it currently "has limited capability to handle contact-rich motion that introduces a large number of discontinuities", which limits its applicability to real-world scenes. It cannot also currently handle joints. All of these would be important for possible robotic applications, for example.

The tasks demonstrated in the experiments are simple, and issues from model mismatch does not seem to have been thoroughly evaluated (see comments below for more).



- Reasons for score

[Edit: Score updated, see discussion below]

Overall, given the "pros" described above, notably the interesting results achieved for system identification and control directly from video frames by combining differentiable physics and rendering into a single framework, I recommend this paper for acceptance. Given some of the concerns raised in the "cons" and in more detail in the comments below, I for now will score this paper as a little above the acceptance threshold.



- Additional comments

The scenarios used for the system identification and control tasks are fairly simple, with usually only a single object and few contact points.
Was the ground truth for the scenarios in the system identification tasks generated using gradsim itself? If so, isn't it unfair that it is compared to other models (e.g., pybullet), for which there would be model mismatch? (While not mismatch would be present for gradsim)

Along the same direction, the experiments present a section on "Impact of imperfect dynamics and rendering models". It would also be interesting to see a quantification of the impact of model mismatch (possibly both while using the same renderer, i.e. only dynamics mismatch, or also different renderers)

In the experiments section, it is said that "Inference ... is done by picking an initial guess of the mass (at random)". From what distribution is this random initial guess picked from? What are these starting guesses in relation to the true parameters?

The section on "Impact of shading and texture cues" seems a little too short, which renders it hard to understand in detail what is going on.

---

> ### Author Response · Authors · 2020-11-24
> **Thank you for the constructive review**
>
> Thank you for your highly constructive review.
>
> We appreciate your suggestions, particularly those concerning additional analyses and ablations. We have since revised our manuscript to address many of these concerns.
>
> **Comment: "Overall, the proposed method is mostly a unification of ... differentiable dynamics, differentiable rendering"**
>
> Response: We acknowledge existing work _independently_ addresses differentiable physics/dynamics and differentiable rendering. However, existing differentiable physics and differentiable rendering methods also have several non-differentiable components (discontinuities), the interactions between which have not yet been fully explored. Surprisingly, we found the unification of these two paradigms results in superior performance compared to state-of-the-art methods, requiring far less supervision (we do not assume access to the true states, e.g.: positions/velocities, and only require image sequences and geometry).
>
> Our performance gain can be explained by comparing the loss landscape in gradSim with that of prevailing methods (e.g. non-differentiable simulators + gradient estimation, such as our PyBullet + REINFORCE baseline, see _Fig. 4_).
>
>
> **Comment: "The paper itself admits .... limited capability to handle contact-rich motion ... cannot handle joints"**
>
> Response: We strongly agree that these and related obstacles are interesting and important avenues for future work. While our contact models (_Section C_) can already handle complex collisions between deformable, cloth and rigid bodies and we have successfully implemented simple articulated bodies such as pendula (as listed in our appendix), we acknowledge that there is a “reality gap” to be overcome for application of gradSim to robotics tasks.
>
>
> **Comment: "The tasks ... are simple, and issues from model mismatch does not seem to have been thoroughly evaluated"**
>
> Response: We have revised our draft to contain a thorough analysis of several model mismatch issues. We have also performed several ablation studies to investigate the impact of modifying dynamics under a fixed differentiable renderer, and the modifying the differentiable renderer under a fixed differentiable dynamics model. We find (as reported in _Sec. 4.3.1_ and _Sec. 4.3.2_) that dynamics cues tend to have a larger effect than rendering cues. We also investigate several other interesting phenomena, such as "what happens if contact goes unmodelled?", and "what if a deformable object is accidentally modeled as a rigid object? (and vice versa)".
>
>
> **Comment: "the scenarios used for the system identification and control tasks are fairly simple ... few contact points"**
>
> Response: Our rationale for using a single object stems from the system identification motivation, where a single object of interest is assumed. Our experiments featuring deformable object and cloth meshes contain 5k -10k vertices, and use all-pairs testing to generate contacts, handled differentiably at each timestep. These meshes are fairly complex and involve tens of thousands of contacts. We agree that system identification in the wild would be an exciting follow-up direction.
>
> **Comment: "Was the ground truth ... generated using gradSim itself? If so, isn't it unfair ..."**
>
> Response: This is a good question and prompted us to add clarification in the paper (please see updated caption for _Fig. 4_). For a fair evaluation, we use rollouts from each simulator to independently define its ground truth (e.g. for the PyBullet + REINFORCE baseline, ground truth rollouts were collected from PyBullet).
>
>
>
>
> **Comment: "Inference ... is done by picking an initial guess of the mass (at random) ..."**
>
> Response: We pick a mass uniformly at random from our operating range of mass densities ($[2, 12] kg/m^3$). While originally featured in the appendix (_Sec. G_), we have now highlighted this point in the main text (_Sec. 4.1.1_). We also emphasize that our loss landscape is smooth for a wide range of initialization errors (cf. _Fig. 4_). We thank the reviewer for raising this issue.
>
> **Comment: "The section on impact of shading and texture cues seems a little too short ..."**
>
>  Response: We agree with the assessment, and have added a more elaborate discussion in _Section 4.3.3_ (including a clarified _Figure 6_ (now _Figure 7_) to address other reviewers’ concerns). For a cleaner demonstration, we exclusively focused on 3 variants, as opposed to 7 in the earlier version.
>
> We hope that our clarification and revision address your primary concerns. We are grateful for your feedback, which played an instrumental role in improving our manuscript.

---

> > ### Comment · AnonReviewer1 · 2020-11-25
> > **Response**
> >
> > Thank you for your thorough response.
> >
> > For the points I had raised in the comments for which I had doubts, your response has clarified these. Moreover, the additions to the paper also greatly improve the paper and help address these points.  Finally, for the few negative points and limitations I have pointed out, I agree that these are natural for a work at this stage. Therefore, given the information contained in the response and the updated paper, I will improve my previous overall assessment of the paper to a "good paper, accept" evaluation.

---

### Author Response · Authors · 2020-11-24
**Revised draft uploaded**

We thank all our reviewers for their thoughtful and constructive feedback. We have revised our manuscript to address most of the comments received. We summarize the major changes here, and respond individually to reviewers to address more specific concerns.

* We have added a thorough explanation of our analysis of **imperfect dynamics and rendering models** (_Section 4.3.1._, _Section 4.3.2._)
* We have revised our section on **imperfect shading cues** (_Section 4.3.3._) with higher resolution figures and a more detailed explanation.
* We have clarified our **choice of baselines** (_Section 4.1.1._) and other design decisions, pointing readers to appropriate locations in the appendix where applicable.
* We added an additional analysis (**impact of video length on performance**) in our appendix (_Section F.3._ and _Fig. 12_)
* We added more details about our **run time** (_Section 4.3.4._ and _Section I_)
* We added an **overview figure** to help better ground our range of applications (_Fig. 1_)

---

### Decision · Program_Chairs · 2021-01-07
**Final Decision**

**Decision:**

Accept (Poster)

**Comment:**

This paper presents a framework for joint differentiable simulation of physics and image formation for inverse problems. It brings together ideas from differentiable physics and differentiable rendering in a compelling framework.